# Interpersonal Sensitivity and Loneliness among Chinese Gay Men: A Cross-Sectional Survey

**DOI:** 10.3390/ijerph16112039

**Published:** 2019-06-08

**Authors:** Dongdong Jiang, Yitan Hou, Xiangfan Chen, Rui Wang, Chang Fu, Baojing Li, Lei Jin, Thomas Lee, Xiaojun Liu

**Affiliations:** 1School of Health Sciences, Wuhan University, 115# Donghu Road, Wuhan 430071, China; dongdjiang@whu.edu.cn (D.J.); houyitan@whu.edu.cn (Y.H.); chenxiangfan@whu.edu.cn (X.C.); elvin_wang@whu.edu.cn (R.W.); baojingli@whu.edu.cn (B.L.); 2Global Health Institute, Wuhan University, 115# Donghu Road, Wuhan 430071, China; 2015103050006@whu.edu.cn; 3Department of Sociology, The Chinese University of Hong Kong, RM 431, Sino Building, Shatin, Hong Kong, China; ljin@cuhk.edu.hk; 4Office of Public Health Studies, University of Hawaii at Manoa, Honolulu, HI 96822, USA; tlee3@hawaii.edu

**Keywords:** Chinese gay men, interpersonal sensitivity, loneliness, influencing factors

## Abstract

To understand the current status of, and factors related to interpersonal sensitivity (IS) and loneliness among Chinese gay men. The Chinese version SCL-90-R was used to evaluate the status of IS, and the short-form UCLA Loneliness scale (ULS-8) was used for assessing loneliness level. Associations between demographics and IS were examined by chi-square tests and multivariable logistic regress analysis. Linear regression was used to assess the correlations between demographic factors and IS and loneliness. Dating practices and venues were summarized by multiple responses. Gay men who screened positive IS was identified in 36%. Age (OR_25–29_ = 8.731, 95% CI 2.296 to 33.139), education level (OR_college_ = 0.037, 95% CI 0.046 to 0.911), being the only-child at home (OR_yes_ = 4.733, 95% CI 2.293 to 9.733), monthly income (OR_>7000_ = 0.228, 95% CI 0.055 to 0.944), numbers of current sexual partners (OR_1_ = 0.285, 95% CI 0.129 to 0.629; OR_2_ = 0.109 95% CI 0.027 to 0.431) were related to IS. IS was also associated with a higher score of ULS-8 (β = 6.903, *p* < 0.001). Other variables associated with the score of ULS-8 included: living in a non-nuclear family (β = 0.998, *p* = 0.020), being a college student (β = −1.556, *p* = 0.044), having a higher monthly income (β for 3000–5000 yuan = −1.177, *p* = 0.045; β for over 7000 yuan = −2.207, *p* = 0.002), having sexual partners (all β < 1, *p* < 0.001), being the only-child (β = 1.393, *p* = 0.005). Nearly half of the sample (46.78%) reported that they looked for dating partners on the Internet or dating apps. IS and loneliness are positively correlated. Our study suggests that more humanistic care and social support should be given to Chinese gay men.

## 1. Introduction

Previous studies suggested that chronic loneliness may cause changes in the cardiovascular, immune, and nervous systems [1]. Additionally, the experience of loneliness can induce continual pain and is highly likely to turn into mental illness or exacerbate the psychological dysfunction [2,3,4]. Loneliness could be manifested through external behaviors, such as aggressive behavior [5], alcohol abuse and suicidal behavior [6]. Rokach et al. found that loneliness was influenced by age and culture background [7]. Segrin et al. proved that the reach of family influence on loneliness was still evident even if when considering more distal family relationships [8]. Interpersonal relationship is the most studied variable affecting loneliness. It is suggested that poor-quality relationships and loneliness were closely related [9], while Cheng et al. also indicated that interpersonal relationships were negatively correlated with loneliness [10]. These studies suggest the possibility of a strong association between interpersonal relationships and loneliness.

There are many types of interpersonal relationships, including interpersonal attachment, interpersonal rejection, and interpersonal sensitivity (IS). As a major measuring indicator of interpersonal relationships, IS refers to the propensity to perceive and elicit criticism and rejection from others [11], which serves as the ability of adaption to social function [12]. It has been well documented that IS is positively associated with depression [13] as well as a central feature of social anxiety disorder [14]. Currently, the association between IS and loneliness is still unclear. Related studies conducted to determine the correlations of IS found that parental over-protection could increase IS [15]. Existing research also indicated that a higher level of IS was associated with the lower quality of life and greater mental distress [16,17]. However, there was no relevant research reporting on the association between IS and loneliness. Additionally, relevant studies were mainly carried out among the young adults [17], school students [9,10], and the elderly [3,16]. Little attention has been paid to sexual minorities groups.

Research about gay men and/or lesbians was mainly conducted in developed countries. Previous studies on same-sex group usually focused on the cause of homosexuality [18], social support [19], sexual behavior [20], and sexual health [21,22]. In recent years, the focus has shifted to better understanding of mental health in the gay men and/or lesbians. However, few contributions exist regarding sex-related research from developing and underdeveloped countries and regions, such as Asia and Africa, because of history, religion and policy. Very little research exists on gay men and/or lesbians in China, with most Chinese researchers focusing on the spread and prevention of sexually transmitted diseases (STDs), such as HIV infection, particularly for those men who have sex with men (MSM) [23,24]. This leaves a huge gap in the literature regarding mental health amongst Chinese gay men.

Traditional Chinese culture highly emphasizes family inheritance and reproduction. Homosexuals, especially gay men may face greater misunderstanding and prejudice [25,26]. Thus, great pressure from society and family may be imposed on Chinese gay men. In addition, Chinese society is a ‘RenQing society’, which means Chinese people highly value interpersonal relationships. It could be hard for Chinese gay men to cope with pressures and discrimination in personal relationship. A survey revealed only 21% of Chinese accepted gay men and/or lesbians [27]. Moreover, China currently has no specific laws or policies to guarantee gay men’s legal rights [28]. Chinese gay men may confront with overwhelming social discrimination, family backlash and a lack of legal protection. We suspect that, under this social dilemma, Chinese gay men may be more likely to suffer from psychological problems [25,29], like IS and loneliness. Therefore, the purposes of this study were to investigate the reality and influencing factors of IS and loneliness among Chinese gay men, and to further examine the relationship between them.

## 2. Materials and Methods

### 2.1. Inclusion and Exclusion Criteria of Participants

Participants were recruited both online and offline. Respondents were recruited according to the following inclusion criteria: (1) males, (2) individuals who voluntarily participated in this survey, (3) self-identified as a gay. The exclusion criteria of potential participants were as follows: (1) male persons who engaged in sexual activity with other men, but did not sexually self-identified as gay, (2) bisexuals, (3) individuals who have been diagnosed with a mental illness. 

### 2.2. Procedures

We utilized a cross-sectional design, and it was conducted from November to December, 2017. We informed the respondents that our survey was conducted anonymously and that the questionnaire did not include respondents’ personal contact information. To increase the diversity of participants, online questionnaire links were sent by web-based live chat applications designed specifically for gay men, such as Blued, Gaypark and Aloha. We also shared survey links to online chat communities for gay men in MoMo, QQ and WeChat. Investigators from the Chinese Center for Disease Control and Prevention (China CDC) conducted the offline survey in their pilot sites. All questionnaires were filled by participants themselves. We adopted a one-on-one online anonymous chat style and self-administered questionnaire combining with a face-to-face interview in official pilot sites, receiving 298 online questionnaires and 78 offline questionnaires, respectively. We conducted a logic error-check inference on the online questionnaires and screened the consistent answers or the blank content to ensure accuracy. We excluded 9 online questionnaire responses, with an effective rate of 96.98%. The offline questionnaires completed by the national CDC were valid and met the inclusion and exclusion criteria. After data collection, we used EpiData 3.1 software (The EpiData Association, Odense, Denmark) to create the database for the offline questionnaires. To make sure the accuracy of the database, we adopted double entry and logical validation.

### 2.3. Instruments

The questionnaire used in this study consists of the following three parts:

#### 2.3.1. Demographic Information

Questions regarding demographics include participants’ age, household registration, education level, family structure, and monthly income. We also asked if the participant is the only-child in the family, and their years of self-identifying sexual orientation, number of current sexual partners and sexual orientation disclosure status.

#### 2.3.2. Measurement of Loneliness

The short-form UCLA Loneliness scale (ULS-8) contains 8 items selected from the revised UCLA Loneliness Scale of Hays and Dimatteo [30,31]. A 4-point Likert scale (1 = never, 2 = seldom, 3 = sometimes, 4 = always) was adopted and two items were reverse-coded prior to analyses. The ULS-8 was confirmed to have good reliability and validity by many scholars [32,33].

#### 2.3.3. Measurement of IS

The SCL-90 intends to measure symptom intensity on nine different subscales, including somatization, obsessive-compulsive, interpersonal sensitivity, depression, anxiety, hostility, phobic anxiety, paranoid ideation, psychoticism. Ninety items of the questionnaire utilize a five-point Likert scale (1 = not at all, 2 = a little bit, 3 = moderately, 4 = quite a bit, 5 = extremely) [34]. The average scores for each item were reported with higher scores of the SCL-90-R indicating greater risk for mental health issues. Many Chinese scholars have proved good reliability and validity of the Chinese version SCL-90-R [35,36,37,38]. Our study adopted the dimension of IS (contains nine items), and the participants’ total average score of IS ≥ 3 was identified as positive.

### 2.4. Statistical Analysis

All analyses were conducted using SPSS software, version 22.0 (SPSS Inc., Chicago, IL, USA), with a significance level of 0.05. Chi-square tests were used to explore the bivariate relationships between demographic factors (age, household registration, educational level, etc.) and IS. Multivariable logistic regression was conducted to analyze the influencing factors between IS and demographic characteristics. And *t*-tests were employed to detect the differences between each item (including total scores) of ULS-8 loneliness scale and IS detection. The multiple linear regression model adjusted for potential confounders, including educational level, family structure, one-child or not at home, monthly income, years of identifying sexual orientation, numbers of current sexual partners, disclose sexual orientation or not. Respondents also selected all possible dating practices and venues.

### 2.5. Ethical Statements

This study was proceeded on the basis of the Declaration of Helsinki. Permission was obtained from the School of Health Science IRB of Wuhan University (MS2017024). The China CDC also reviewed this study, and offered great help in the offline data collection process.

## 3. Results

### 3.1. Descriptions of Sample Characteristics

Descriptive statistics for all measures are presented according to IS screening status in Table 1. A total of 131 participants (35.69% of the total sample) tested positive for IS. Chi-square tests illustrated that age (χ^2^ = 54.653, *p* < 0.001), educational level (χ^2^ = 29.118, *p* < 0.001), being the only-child at home (χ^2^ = 99.941, *p* < 0.001), monthly income (χ^2^ = 62.552, *p* < 0.001), current sexual partner numbers (χ^2^ = 69.885, *p* < 0.001) and situation of opening sexual orientation (χ^2^ = 75.155, *p* < 0.001) were significantly associated with IS.

### 3.2. The Factors Affecting IS

Table 2 lists the adjusted odds ratios (ORs) obtained from multivariable logistic regression model with the 95% confidence intervals (CIs). The results showed that gay men who aged 25–29 were more likely to present with IS, as compared with gay men who were under 20 years old (OR = 8.731, CI: 2.296–33.199). Gay men who had a college degree (OR = 0.204, CI: 0.046–0.911) were less likely to be detected as positive for IS, as compared with those who had junior high school education or lower. We also found that gay men who were the only-child at home had a higher risk in IS (OR = 4.733, CI: 2.293–9.773). When compared with those whose monthly incomes were less than 3000 yuan, gay men with a monthly income over 7000 yuan showed lower possibilities in having IS (OR = 0.228, CI: 0.055–0.944). In addition, gay men having at least one sexual partner were also less likely to be detected as positive for IS.

### 3.3. Scores of Total Loneliness and Its Eight Items

Table 3 recorded that all ULS-8 items were associated with positive rate of IS. Each item’s score and total scores of those who screened positive for IS were higher than those who screened negative. Total ULS-8 scores of subjects who screened negative and positive for IS in loneliness were 15.08 and 25.45, respectively.

### 3.4. Factors Associated with Loneliness, and the Relationship of Loneliness and IS

Similar with the unadjusted results showed in Table 4, the adjusted results displayed in Table 4 identified that participants who were screened positive for IS showed a higher score in loneliness (β = 6.903, S.E = 0.537, *p* < 0.001). 

Among the demographic characteristic factors, being the only-child (yes, β = 1.393, S.E = 0.490, *p* = 0.005) and family structure (others, β = 0.998, S.E = 0.425, *p* = 0.020) were positively associated with the loneliness. Whereas the education level (college, β = −1.556, S.E = 0.769, *p* = 0.044), monthly income (3001–5000, β = −1.177, S.E = 0.585, *p* = 0.045; > 7000, β = −2.207, S.E = 0.722, *p* = 0.002), numbers of current sexual partners (1, β = −2.852, S.E = 0.518, *p* < 0.001; 2, β = −2.075, S.E = 0.648, *p* = 0.001; ≥3, β = −2.276, S.E = 0.626, *p* < 0.001), and situation of disclosing the sexual orientation (open, β = −1.637, S.E = 0.505, *p* = 0.001) were negatively associated with the loneliness.

### 3.5. Measurement of Dating Practices and Is Detection Amongst Chinese Gay Men

The dating practices of Chinese gay men were presented in Table 5 and Figure 1. Almost half of the responses (46.78%) used the internet or dating apps, of whom 115 participants were screened positive in IS. Gay bar or dance hall was the second commonly selected venue. Only 6.45% of the responses indicated that their frequent meeting places were at bathhouses. The proportion of gay men who detected positive for IS in internet/dating app, gay bars/dance halls, tea house/clubs, bathhouses, parks/toilets/lawns, others was 36.86%, 47.52%, 48.75%, 44.18%, 46.55%, 38.36%, respectively.

## 4. Discussion

Gay is still a sensitive topic in China, and many Chinese, especially the elderly, look down upon gay men because of the deep-rooted traditional morals that overemphasize fertility and patriarchy [29], causing mental health issues in Chinese gay men. The current research attempts to build a bridge between natural science and social science by providing a baseline understanding of mental health issues in Chinese gay men.

This study revealed that gay men who aged 25–29 had higher positive rate of IS, which is in line with a previous study that declared mood-related IS in younger ages was more common across the lifespan [39]. People aged 25–29 in China usually have joined the workforce, considering young sexual minorities have a greater risk of experiencing continuous discrimination, violence and rejection [40], and relationship at work is positively associated with mental health [41], thus gay men may be afraid of disclosing their sexual orientation which could lead to IS. Educational background is widely seen as a major indicator of measuring mental health, which was also confirmed by our study. Gay men with college degree are less likely to be detected positive in IS. Tong et al. suggested that people with junior high school education or lower have higher scores in IS than those with undergraduate degrees [42], indicating that those with higher education may have a decreased chance to be susceptible to IS. In our study, we also found that gay men in China with college level felt less lonely than those with junior high school degree or lower. It is possible that individuals with higher education are less likely to suffer from mental health problems [29]. People who receive higher education may have greater knowledge and skills to handle IS and other mental problems.

Our study reported that gay men who were the only child at home were more likely to be detected positive in IS, as compared with gay men with siblings. This may be explained by the conflicts between the pressures of Chinese traditional filial piety and disclosure of sexual orientation. Furthermore, our study indicated that gay men who were the only child at home were detected higher levels of loneliness. Fu et al. believed an obvious difference existed in mental health between twins and only-child [43]. Gay men who are the only child may face even more pressure. They may fear to disclose their sexual orientation. Ryan et al. suggested that it was risky for gay men to disclose their sexual orientation because of prejudice and family rejection [44]. In the worst cases, rejection from family may result in the risk of suicide and substance misuse [45]. Given that the one-child policy in China was implemented nearly 40 years ago, the number of people who are gay men among the only- child families is considerable. It is necessary to increase related public education to make the society, especially family members, understand and accept this vulnerable group [28]. This would reduce the pressure on gay men, allowing them to face their sexual orientation and avoid the risk of disclosing sexual orientation to their family.

In our research samples, the number of gay men who kept their sexual orientation confidential was almost 7 times than that of those who openly shared their sexual orientation. Research conducted by San Francisco State University revealed that family rejection was significantly associated with poorer health outcomes for LGB young adults [46]. It is generally agreed that disclosing sexual orientation to others is beneficial for gay men and their relationships, but most parents tend to react with shock, disappointment and shame [47]. Disclosure of sexual orientation is most likely to result in a family crisis and create rifts between family members [48]. We did not find whether open sexual orientation or keep it confidential was associated with IS, but those who wholly open their sexual orientation felt less loneliness than those who keep it confidential. 

The majority (80%) dating practices of Chinese gay men are done through the Internet. We also found that gay men who detected positive in IS chose online dating, accounting for a fewer proportion than offline dating. We speculate that it is because the internet allows anonymity during the early communication process, which could make gay men reduce the possibility of rejection in making virtual friends and get comfort in the virtual world. A current study indicated that online interaction could fill a void in the offline world and play an important role in the daily lives of people who live with HIV/AIDS [49]. This situation can also apply to gay men. Therefore, it is necessary for the society to increase acceptance and build more public venues to reduce the isolation of Chinese gay men [50]. The relevant government departments, especially public health institutions, need to provide financial support and offer counseling to raise Chinese gay men’s psychologically healthy level.

Previous studies confirmed that adolescents from single parent and blended families were more likely to be lonely and had worse health status compared with adolescents from intact families [51,52]. Our study demonstrated that risks of experiencing loneliness in non-nuclear family structures (one-parent, blended or united families) were higher than nuclear families among Chinese gay men. It is a common phenomenon that Chinese adults aged over 22 years old have reached the legal age of marriage, and their parents may urge them to get married as soon as possible. It could be even worse for adolescents from non-nuclear family, for they may face pressure from both their parents and brothers and/or sisters.

We found that gay men who earn over 7000 yuan had lower IS positive rate. Esmina et al. noted that socio-economic factors such as the level of income could be the predictors of psychological symptoms [53]. We assume that high income may meet gay men’s material possessions and lead to a more equal identity to talk about their sexual orientation. In line with previous findings [54,55], loneliness among Chinese gay men in our study was reduced with higher personal monthly income. Our paper also revealed that IS among Chinese gay men was significantly affected by the number of current sexual partners. To some degree, gay men are always in need of finding sexual partners [56]. Sexual behaviors can increase emotional contact and alleviate insecurity [57], which may relieve their IS. In addition, sexual partners can also decrease Chinese gay men’s loneliness. As our research has shown, few gay men were willing to open their sexual orientation. But sex partners can provide access to converse, which may reduce the level of loneliness.

Our study reported that gay men with positive detection in IS felt much more loneliness than those with negative detection. Butler et al. found that IS and interpersonal skills were negatively correlated [58]. Duygu et al. and Mccabe et al. also reported that IS was related to negative coping styles [59,60]. Gay men with high IS may have difficulties in dealing with interpersonal communication and give some negative feedback to people around them, because they do not want their sexual orientation to be found. Previous studies have reported that IS was a susceptible factor for depression and anxiety [13,14], while depression and anxiety were also closely related to loneliness [61]. Therefore, it is essential to encourage Chinese gay men to conduct psychological counseling when they realize the symptoms of these mental illnesses. Social media should help to increase the dissemination of sex-related knowledge and public acceptance to Chinese gay men, which could decrease the risk of STDs and protect the right of gay men [62].

As the first study addressing IS and loneliness among gay men in mainland China, the following limitations of this study should be noted. First, we used non-random sampling, including convenient sampling and snowball sampling, which may lead to some bias. Moreover, finding enough gay men samples is quite difficult, because most gay men in China are used to hiding their sexual orientation in order to avoid social discrimination, stigma and pressure from family [25,26]. In fact, many of them refused to participate, and most gay men might just ignore our online messages. Only those agreed to take part in our survey were included in the analysis. It is possible that those who completely voluntarily involved in this study may be more open-minded and with a better mental state. Therefore, the research outcome of Chinese gay men’s IS and loneliness could be underestimated. Sending survey links online can limit access to certain sample populations, for only those who use the related apps and surf the Internet could join this survey. Results showed most participants were young and received a certain education. Finally, the non-response rate was not able to assess. Thus, the results of this study should be cited with caution.

## 5. Conclusions

This study first clearly reveals that IS and loneliness are positively correlated in Chinese gay men. We also found gay men who aged 25–29 and are the only-child at home could be more likely to be detected IS positive. College degree, monthly income over 7000 yuan and sexual partners are the protective factors of decreasing IS positive rates. The factors lead to loneliness for Chinese gay men are living in a non-nuclear family, being the only-child at home. While being a college student, having a higher monthly income, having sexual partners, opening sexual orientation can reduce the risk of loneliness of Chinese gay men. Results of this paper suggest that we need to be more aware of the Chinese gay men’s mental health, especially their feelings of IS and loneliness. To minimize the level of IS and loneliness, actions should be taken in the care for the Chinese gay men. The government should encourage everyone, especially family members, to give more support and humanistic care to Chinese gay men. The social environment should be more open and inclusive. Psychological counseling centers should be established to provide mental health evaluation. More dating sites should be built to increase the chance to attend group communication.

## Figures and Tables

**Figure 1 ijerph-16-02039-f001:**
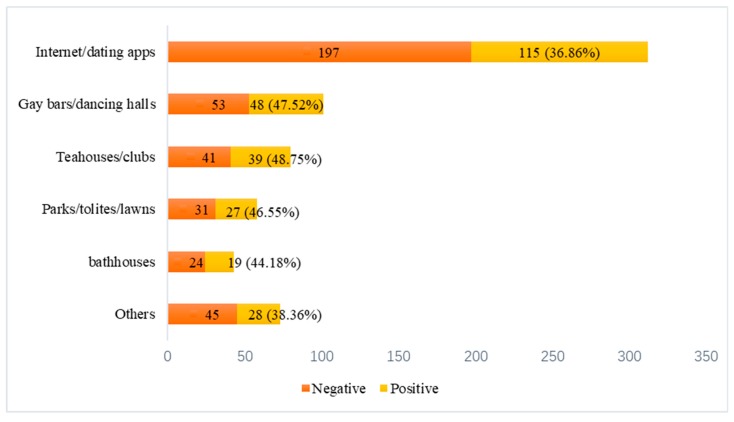
Dating venues and IS detection among Chinese gay men.

**Table 1 ijerph-16-02039-t001:** Demographic information of participants by interpersonal sensitivity status (*n* = 367).

Demographics	Negative	Positive	χ^2^	*p*-Value
	*n* = 236 (64.31%)	*n* = 131 (35.69%)		
Age			54.653	<0.001
<20	56 (76.71)	17 (23.29)		
20–24	54 (72.00)	21 (28.00)		
25–29	31 (35.63)	56 (64.37)		
30–34	29 (53.70)	25 (46.30)		
≥35	66 (84.62)	12 (15.38)		
Household registration			0.860	0.354
Countryside	107 (61.85)	66 (38.15)		
City	129 (66.49)	65 (33.51)		
Educational level			29.118	<0.001
Junior high school and lower	12 (33.33)	24 (66.67)		
High school	55 (53.92)	47 (46.08)		
College	138 (73.02)	51 (26.98)		
Post-graduate and higher	31 (77.50)	9 (22.50)		
Family structure			0.035	0.851
Nuclear family	159 (64.63)	87 (35.37)		
Others	77 (63.64)	44 (36.36)		
Being the only-child at home			99.941	<0.001
No	178 (86.41)	28 (13.59)		
Yes	58 (36.02)	103 (63.98)		
Monthly income (RMB)			62.552	<0.001
<3000	99 (70.71)	41 (29.39)		
3001–5000	30 (32.61)	62 (67.39)		
5001–7000	42 (67.74)	20 (32.26)		
>7000	65 (89.04)	8 (10.96)		
Years of identifying sexual orientation			2.941	0.401
≤3	58 (57.43)	43 (42.57)		
4–6	49 (66.22)	25 (33.78)		
7–9	43 (66.15)	22 (33.85)		
≥10	86 (67.72)	41 (32.28)		
Numbers of current sexual partners			69.885	<0.001
0	59 (40.97)	85 (59.03)		
1	90 (81.08)	21 (18.92)		
2	52 (94.55)	3 (5.45)		
≥3	35 (61.40)	22 (38.60)		
Disclose sexual orientation or not			75.155	<0.001
Confidential	95 (45.45)	114 (54.55)		
Open	141 (89.24)	17 (10.76)		

Note: RMB 3000, 5000, and 7000 equal about USD 434, 724, and 1013, respectively using an exchange rate of USD to RMB 1 to 6.91.

**Table 2 ijerph-16-02039-t002:** Logistic regression analysis of influencing factors of interpersonal sensitivity.

Variables (Control Group)	β	S.E	*p*-Value	OR	95% CI for OR
Age (<20)					
20–24	0.511	0.495	0.303	1.666	0.631–4.399
25–29	2.167	0.681	0.001	8.731	2.296–33.199
30–34	1.000	0.773	0.195	2.719	0.598–12.369
≥35	−1.543	0.801	0.054	0.214	0.044–1.028
Household registration (Countryside)					
City	−0.509	0.373	0.173	0.601	0.289–1.250
Educational level (Junior high school and lower)					
High school	−1.048	0.692	0.130	0.351	0.090–1.361
College	−1.588	0.762	0.037	0.204	0.046–0.911
Post-graduate and higher	−1.700	0.949	0.073	0.183	0.028–1.173
Family structure (Nuclear family)					
Others	0.196	0.375	0.601	1.216	0.583–2.536
Being the only-child at home (No)					
Yes	1.555	0.370	<0.001	4.733	2.293–9.773
Monthly income (RMB) (<3000)					
3001–5000	0.718	0.473	0.129	2.050	0.811–5.179
5001–7000	−0.958	0.604	0.113	0.384	0.117–1.254
>7000	−1.479	0.725	0.041	0.228	0.055–0.944
Years of identifying sexual orientation (≤3)					
4–6	−0.672	0.477	0.159	0.511	0.201–1.301
7–9	−0.885	0.638	0.166	0.413	0.118–1.442
≥10	−0.145	0.626	0.817	0.865	0.253–2.953
Numbers of current sexual partners (0)					
1	−1.257	0.405	0.002	0.285	0.129–0.629
2	−2.219	0.702	0.002	0.109	0.027–0.431
≥3	0.059	0.530	0.912	1.060	0.375–2.996
Disclose sexual orientation or not (Confidential)					
Open	−0.778	0.411	0.058	0.459	0.205–1.027

Note: RMB 3000, 5000, and 7000 equal about USD 434, 724, and 1013, respectively using an exchange rate of USD to RMB 1 to 6.91; β = Coefficient; S.E = Standard Error; OR = Odds Ratio; CI = Confidence Interval.

**Table 3 ijerph-16-02039-t003:** Assessment results of each item of ULS-8 ($\bar{x}$ ± s).

Items of ULS-8	Negative	Positive	*t*/*t*’	*p*-Value
I lack companionship.	2.10 ± 0.82	3.50 ± 0.65	−16.70	<0.001
There is no one I can turn to.	1.83 ± 0.87	2.95 ± 0.66	−13.78	<0.001
I feel left out.	1.92 ± 0.81	3.37 ± 0.67	−17.41	<0.001
I feel isolated from others.	1.89 ± 0.82	3.35 ± 0.64	−17.64	<0.001
I am unhappy being so withdrawn.	1.84 ± 0.82	3.29 ± 0.72	−16.97	<0.001
People are around me but not with me.	1.79 ± 0.78	3.34 ± 0.60	−21.22	<0.001
I am an outgoing person.	1.75 ± 0.93	2.78 ± 0.71	−11.93	<0.001
I can find companionship when I want it.	1.95 ± 0.94	2.89 ± 0.58	−11.78	<0.001
Total scores	15.08 ± 4.37	25.45 ± 3.77	−23.83	<0.001

**Table 4 ijerph-16-02039-t004:** Estimated adjusted associations (β and 95% CI) of loneliness by interpersonal sensitivity and other factorss.

Variables	Unstandardized Coefficients		Standardized Coefficients	*t*	*p*-Value	95% CI for β	Collinearity Statistics	
	β	S.E	Beta				Tolerance	VIF
Interpersonal sensitivity (negative = control group)	6.903	0.537	0.511	12.866	<0.001	(5.848, 7.959)	0.528	1.894
Education level (≤Junior high school = control group)								
High school	−0.969	0.730	−0.067	−1.327	0.185	(−2.405, 0.467)	0.326	3.067
College	−1.556	0.769	−0.120	−2.024	0.044	(−3.068, −0.044)	0.236	4.232
≥Post-graduate	−1.607	0.982	−0.077	−1.637	0.103	(−3.537, 0.324)	0.373	2.683
Family structure (nuclear family = control group)	0.998	0.425	0.072	2.346	0.020	(0.161, 1.834)	0.873	1.146
Being the only-child at home (No = control group)	1.393	0.490	0.107	2.847	0.005	(0.431, 2.356)	0.591	1.692
Monthly income (RMB) (≤3000 = control group)								
3001–5000	−1.177	0.585	−0.079	−2.014	0.045	(−2.327, −0.028)	0.543	1.840
5001–7000	−0.161	0.658	−0.009	−0.245	0.807	(−1.454, 1.132)	0.575	1.740
>7000	−2.207	0.722	−0.136	−3.058	0.002	(−3.627, −0.788)	0.420	2.380
Numbers of current sexual partners (0 = control group)								
1	−2.852	0.518	−0.202	−5.508	<0.001	(−3.870, −1.834)	0.617	1.621
2	−2.075	0.648	−0.114	−3.201	0.001	(−3.350, −0.800)	0.652	1.534
≥3	−2.276	0.626	−0.127	−3.638	<0.001	(−3.506, −1.045)	0.680	1.472
Disclose the sexual orientation or not (confidential = control group)	−1.637	0.505	−0.125	−3.242	0.001	(−2.630, −0.644)	0.558	1.791
F = 42.834, R^2^ = 0.696, 0 < *p* < 0.001								

Note: RMB 3000, 5000, and 7000 equal about USD 434, 724, and 1013, respectively using an exchange rate of USD to RMB 1 to 6.91.; β = Coefficient; S.E = Standard Error

**Table 5 ijerph-16-02039-t005:** Multiple responses of dating venues or ways to Chinese gay men.

Dating Venues or Ways	Responses		Percent of Cases
	Total	%	
Internet/Dating Apps	312	46.78	85.01
Gay Bars/Dance Halls	101	15.14	27.52
Tea Houses/Clubs	80	11.99	21.80
Bathhouses	43	6.45	11.72
Parks/Toilets/Lawns	58	8.70	15.80
Others	73	10.94	19.89
Total	667	100	181.74

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
