# Peer review of "Interpersonal Sensitivity and Loneliness among Chinese Gay Men: A Cross-Sectional Survey"

_ijerph, 2019, doi:10.3390/ijerph16112039_

Round 1
Reviewer 1 Report
I enjoyed reading this very novel and groundbreaking journal article. The paper was easy to read and the study is easily replicated since the authors were very thorough in describing their method. While the results clearly indicate that Chinese men continue to live in hostile environments, the brief conclusions show that there is much room for further study on how governments and family units can propel this group forward.
Author Response
I enjoyed reading this very novel and groundbreaking journal article. The paper was easy to read and the study is easily replicated since the authors were very thorough in describing their method. While the results clearly indicate that Chinese gay men continue to live in hostile environments, the brief conclusions show that there is much room for further study on how governments and family units can propel this group forward.
Response: We thank the reviewer for this supportive comment.

Reviewer 2 Report
The term “homosexuals” should be replaced by “gay men and lesbians.”
Abstract
1. ULS-8: full spelling
2. Dating practices and venues were not related to the aim of the present study and should be deleted.
Introduction
1. The aims of this study should be described.
2. “We suspect that, under this social dilemma, Chinese gay men may be more likely to suffer from psychological problems, like IS and loneliness.” Its seems like that the authors plan to examine whether Chinese gay men may be more likely to suffer from IS and loneliness than heterosexuals. But not heterosexuals were recruited into this study.
Methods
1. Participants were recruited both online and offline. Is there any difference between participants recruited online and offline?
2. “Our study adopted the dimension of interpersonal sensitivity and identified each item’s score ≥ 3 as positive detection in interpersonal sensitivity.” Please clarify more. Any item or all items ≥ 3?
3. “Chi-square tests were used to explore the bivariate relationships between demographic factors (age, household registration, educational level, etc.) and IS.” Further multivariate logistic regression analysis is necessary. Several factors mau correlate with each other.
4. How about the bivariate relationships between demographic factors and loneliness?
5. “T-tests were employed to detect the differences between each item of ULS-8 loneliness scale and IS detection.” What is the purpose to examine “each item of ULS-8” but not the total score?
6. “The multiple linear regression model adjusted for potential confounders.” What are dependent and independent variables? Why did the authors select IS as the independent variable and loneliness as the dependent variable?
Results
1. Thecontents should be revised based on the new results of statistical analysis mentioned above.
2. Dating practices and venues were not related to the aim of the present study and should be deleted.
Discussion
The contents should be revised based on the new results of statistical analysis mentioned above.
Author Response
Dear Reviewer,
We are respectfully submitting a revised version of our manuscript, entitled “The Influencing Factors of Interpersonal Sensitivity and Loneliness among Chinese Gay Men”, for your consideration for publication in the Special Issue "Health and Wellbeing in Sexual Orientation and Gender Identity" of International Journal of Environmental Research and Public Health.
The authors appreciate the thoughtful and critical feedback from the reviewers very much. We are delighted at your decision. The manuscript has been revised according to three reviewers’ comments, and all changes have been highlighted for ready identification, and our responses have been outlined in a comment/response format below (see responses to the comments).
Hope the revision is satisfactory and this manuscript is now acceptable for publication in your journal.
I am looking forward to hearing from you soon.
Best,
Jim

Reviewer 3 Report
Dear Autors,
I believe that this work is well done especially concerning the methods and tables. I also believe that the conclusions must be increased by least one two lines.
Thank you.
Best regards.
Author Response
Dear Autors,
I believe that this work is well done especially concerning the methods and tables. I also believe that the conclusions must be increased by least one two lines.
Thank you.
Best regards.
Response: We thank the reviewer for the supportive and valuable review comments. We have re-organized the conclusions as suggested.

Round 2
Reviewer 2 Report
The authors have revised their manuscript adequately. I would like to suggest the editors accepting this manuscript for publication.
Author Response
The authors very much appreciate the thoughtful and critical feedback from the reviewer, and we are delighted at your decision.
Thank you.